# Proximal Sensing of Nitrogen Needs by Spring Wheat

Shlomo Sarig [1], Eli Shlevin [1], Arkadi Zilberman [2], Idan Richker [3], Mordechay Dudai [1], Shlomo Nezer [1] and Jiftah Ben-Asher [2,*] 

1   Katif R&D Center, Ministry of Science, Jerusalem 8771002, Israel; shlomosarig@gmail.com (S.S.); eshlevin@saad.org.il (E.S.); mordudai@gmail.com (M.D.); momi1805@gmail.com (S.N.)
2   AgrIOT Group, Ben-Gurion University of the Negev, Be'er Sheva 8410501, Israel; arkadiz@gmail.com
3   Field Crops Organization, Eilot Region 8882000, Israel; idanrich@gmail.com
*   Correspondence: benasher@bgu.ac.il

**Abstract:** Canopy nitrogen (N) status relates strongly to canopy chlorophyll content and the strength of green color. Proximal photograph by RGB camera was used to select green features that has the potential to assess N content at leaf of plant as a function of its the greenness. We proposed the development of it as a tool for sensing nitrogen content in spring wheat (*Triticum aestivum*). Image processing algorithm was programed calibrated and validated wheat %N%N. Nitrogen uptake =%N × canopy dry matter was harvested and calculated using simulated dry matter by DSSAT model. The data replicated laboratory measurements. A linear Lab vs Camera model displayed a unit slope with $r^2 = 0.93$. Increase of dry matter was successfully surrogated by days after emergence and used as abscissa for inverse logistic model of critical nitrogen level. It decreased gradually from about 6% to 2% as days after emergence increased from 0 to 110 days. Maximum N uptake calculated from photo and laboratory was 324 Kg ha$^{-1}$ and 318 Kg ha$^{-1}$ respectively suggesting insignificant difference. Physiological N-use efficiency (i.e., canopy weight/N weight) was 52 and 78 kg canopy dry weight per 1 kg N for early and late-ripening cultivars, respectively. The determination of N application based on the smartphone photograph proved to be useful by saving on time and expenses for growers who have access to smartphones and can use them for N application and management.

**Keywords:** decision support; fertilization; mobile phone; NNI; critical nitrogen (Nc)

## 1. Introduction

### 1.1. Wheat: Statistical Data

Wheat is grown yearly on 215 million hectares—an area that cover countries or provinces in most continents. It is distributed from Scandinavia in the Northern hemisphere to South America in the southern hemisphere and across Asia in the far east. China alone provides about 10% of this area. The other 88 countries make it more a widely grown than any other food crop. Production varied from 2.75 t/ha. to 4.4 t/ha. The national average wheat yield in China is 4.8 tons per ha (t/ha), well above the global yield. China produces more than 117 million tons annually, representing 17% of world wheat output over the same period. The global per capita food use of wheat stands at 66.9 kilograms per year.

Summary of data collected from Google including the statistic provided by the Food and Agricultural Organization (FAO of the United Nations, Rome, Italy).

### 1.2. Proximal Sensing of Nitrogen

Reliable soil nitrogen (N) management continues to be an important challenge in agricultural research and for the economy. Methods to determine quantitative relationships between crop N uptake and yield are continually being sought. Aside from the economic importance of optimizing N use, uncontrolled N supply may lead to severe environmental pollution of drinking water, aquifers, lakes and rivers.

Numerous leaf-level studies with remote sensing techniques have demonstrated a strong link between nitrogen (N) and chlorophyll. Among the studies it is worth noting the work of Gitelson and coauthors [1–3].

In the present work, an applicability of digital color imaging to monitor nitrogen uptake by crops is demonstrated.

The technical methodology and the use of machine learning with artificial neural network was described in detail in proceeding papers [4,5]. This point is emphasized strongly in Section 1.3.

Previously measurements were performed in a carrot, lettuce wheat fields, greenhouse and several other crops [4,5]—the images of canopy were taken during the entire growing season by use a hand held digital color (RGB) camera together with plants collection for lab analysis. The nitrogen weight in plant leaves, Kg ha$^{-1}$ was obtained by image processing and compared with the standard laboratory analysis.

The leaves appear green because the green light band around 550 nm is reflected relatively efficiently when compared with the blue, yellow and red bands, which are absorbed by photoactive pigments. From approximately 670 nm, reflectance/absorbance changes cause the sharp transition from low visible reflectance to high Near Infrared Reflectance (NIR) reflectance.

### 1.3. Image Processing, Machine Learning and Neural Network Analysis

Photo 1 illustrates the approach for assessing N uptake in plant leaves. The approach combines Computer Vision and Machine Learning techniques with Artificial Neural Network (ANN) for real-time monitoring of %N%N. The process starts with an original photo or Input (I) followed by (II) —Preprocessing & Features extraction; (III) —a black box containing a large number of layers of deep learning with ANN and (IV) Output, the Distribution of Nitrogen percent in wheat leaves.

Four stages for assessing N uptake in plant leaves are displayed in Figure 1.

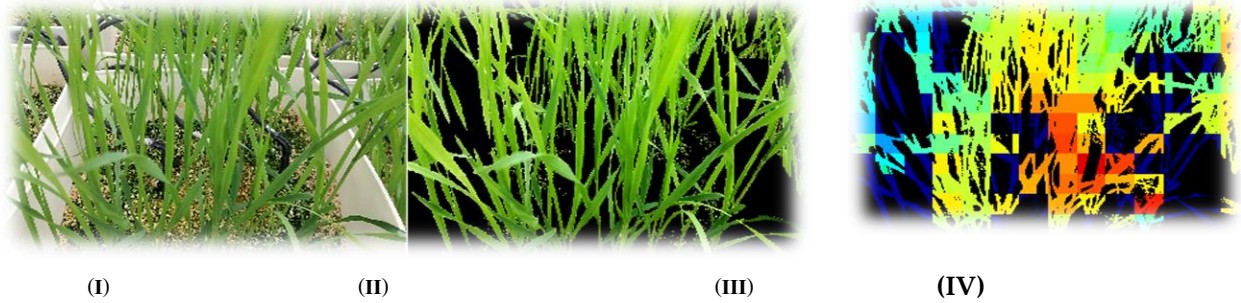

**(I)**      **(II)**      **(III)**      **(IV)**

**Figure 1.** Wheat in greenhouse: (**I**) original photo, (**II**) after preprocessing, (**III**) The empty space represents a large number of layers each layer contain a machine learning with Artificial Neural Network; and (**IV**) N percent map. A scale that includes artificial colors. The scale of the artificial colors is a gradual change in %N from low blue to high red and the average of the field ± its standard deviation. Schematically: Blue goes from 0–2%, yellow 2–3.5%, red = 3.5–5% (average 3.1 ± 0.6). Treatment received N amount corresponding to 120 kg/ha.

### 1.4. Traditional Identification of Preplanting N Potential

Chemical-biological extraction methods have been developed to estimate soil N contents, using wheat or maize as test plants [6].

Throughout wheat's growing period, there are at least three important stages during which N application in topdressing can be effective, while the plant is transitioning from vegetative to reproductive growth. These stages are: (i) 3- to 5-leaf stage, (ii) 6- to 7-leaf stage and (iii) appearance of the flag leaf. Until now, the most reliable means of monitoring a crop's nutritional condition has been to run plant leaf samples in the laboratory. The limited period of the 3- to 5-leaf stage is crucial for wheat N-fertilization decisions. During this period, NO$_3$-N concentration in the petioles is high and easy to measure in the laboratory.

Israeli extension agents adopted it as a good indication of the plant's nutritional needs [7]. However, due to the prolonged sample-processing period, this stage of plant growth is frequently missed in plants sent to the laboratory. A practical alternative was found in the Horiba Cardy nitrate $NO_3^-$ meter sensor, which requires a very short sample preparation and measurement time [8]. The proximal sensing using a digital camera for total N uptake versus days after emergence (DAE) is a promising alternative to the methods described above [9].

### 1.5. Yield Model, Critical Nitrogen (Nc) and Nitrogen Nutritional Index (NNI)

The change in the N percentage (%N%N) with increasing crop biomass is not linear. This phenomenon is common to many crops and it can be expressed by the N-dilution curve:

$$\%N\%N = a \times DM^{-b} \tag{1}$$

where: DM (dry matter) is the crop's dry biomass on the date of %N measurement, and correlation parameters. This concept reveals a connection between crop weight and its N absorption [10].

Moreover, according to Equation (1), when DM = 0, %N increases exponentially to infinity. In other words, there is a point of singularity close to DM = 0, and we therefore modified Equation (1) with some objective corrections for a critical N-dilution (Nc) curve.

The critical N (Nc) concentration is defined as the minimum N concentration required for maximum crop growth ([11–13]). Justes [14] proposed a statistical approach to determining the Nc of field crops. This term is a fundamental reference at any growth stage, enabling a determination of whether the crop's N nutrition is supraoptimal or suboptimal. The $N_{actual}$/Nc ratio is defined as the N nutrition index (NNI) [10,15]. when NNI = 1, N nutrition is considered optimal, at NNI > 1, it is supraoptimal, and at NNI < 1, it is insufficient.

Proximal sensing of plant-based diagnostic technique can be used to determine Nc and NNI in order to support fertilization decisions. Our objectives were to establish the relationship between simple photographic camera readings %N and NNI values during the wheat growing season. In general, the greenness of the color photograph indicating N concentration and NNI improvement with increasing N rates.

### 1.6. Precision Nitrogen Management of Wheat

Previous sections reviewed general Nitrogen management while this section is a specific review on nitrogen management of wheat [16]. Unfortunately, most of the plant-based analyses mentioned above are generally time-consuming, and only a few plants can be sampled inaccurately representing spatial variability of a field. Theoretically assuming a homogeneous field grain yield can change with applied N from about 1.2 Mgha$^{-1}$at 0 N application to 4.5 Mgha$^{-1}$. at 120 Kg N ha$^{-1}$. In extreme cases above average production 5. Mgha$^{-1}$. grain yield can be obtained for a cost of 220 Kg N ha$^{-1}$. Therefore, Nitrogen trials paralleled by economic assessments (cost-benefit) are most needed. Theoretical and experimental optimization study of fertilization that was carried out by Tabak et al. [16] enabled to obtain a yield of high quality and quantity which resulted in economic profits, and reduced environmental threats. They tested three Nitrogen doses 150, 200, and 250 Kg N ha$^{-1}$ from which mean optimal nitrogen was about 217 Kg ha$^{-1}$. and the yield was 8251 Kg ha$^{-1}$ with a physiological efficiency 0f 38 Kg grain for each 1 kg Nitrogen (assuming that for 0 N application there was also 0 yield).

While the report of Tabak et al. [16] implicitly assumed homogenous field, Diacono [17] considered explicitly heterogeneous wheat field. He suggested quantification methods to address the spatial variability of the field by adopting precision agriculture technologies. Among the methods to implement precision agriculture he suggested to using airborne images remote and proximal sensing for predicting crop N status. It was found [17] that field studies in which sensor-based N management systems were compared with common farmer practices showed high increases in the N use efficiency of up to 368%. These systems

saved N fertilizers, from 10% to about 80% less N, and reduced residual N in the soil by 30–50%, without either reducing yields or influencing grain quality.

*1.7. Objectives*
1.7.1. The General Goal

This research combines under one roof (a smartphone application) algorithm to calculate N% in spring wheat. on a basis of accumulated knowledge [10,12,15], simulation model and sophisticated computing capabilities. that is using smartphones to produce supportive recommendation for agronomists and growers.

1.7.2. Specific Objectives

(A) Use an application of smartphone photos to calculate a deterministic dose of nitrogen fertilizer topdressing on spring wheat.
(B) To compare the calculated N uptake by proximal sensing output with laboratory measurements.
(C) To determine a new model for calculating critical nitrogen (Nc) on a basis of days after emergence (DAE) in order to reduce the time and labor requirements for canopy harvest.
(D) To combine NNI and Nc levels for optimal nutritional management of spring wheat.

## 2. Materials and Methods

*2.1. Smartphone Photographic Determination of Wheat Nitrogen Status*

A smartphone camera measuring RGB wavebands of 400 to 700 nm was used to determine the spectral reflectance of wheat. Determination of %N was taken at four stages of growth and based on the cereal code of Zadox [18]. (i) 3 to 4 leaves = GS 14, (ii) 6 to 7 leaves, = GS 17 (iii) flag leaf, = GS 41 and (iv) heading = GS 50. About 100 laboratory tests of %N were taken for calibration and establishment of accuracy standard values. As a standard routine the calibration was followed by a validation process that included 20–30 independent samples. Both stages (calibration and validation) were performed through three comparison stages with laboratory measurements: (I) Preprocessing and features extraction. In this stage the soil pixels were eliminated from the original image analysis by converting RGB to HSV (hue, saturation, value) (II) —machine learning with ANN by a software available in MATLAB. (III) Output of distribution of Nitrogen uptake on the leaves or in the field. It was then compared to laboratory results [5]. The camera was held at variable heights to ensure resolution of 0.02 m per pixel. The photographs were taken when solar radiation was between 1.6 and 1.4 MJ m$^{-2}$ from 1000 to 1400 h in December, and from 3.5 to 3.2 MJ m$^{-2}$ for the same hours in April. The advantage of this optical setup was that no detector other than the smartphone was required. A smartphone is readily available almost to anybody everywhere. Additional is that the sensor is the plant and the smartphone is the data logger that covers a large group of plants within a relatively larger area., averaging results of more samples than a single sample often taken to the laboratory.

*2.2. Greenhouse Lysimeters Trials*

Two experiments, conducted in 2017 and 2018, included various N rates (0–180 Kg ha$^{-1}$) and two wheat cultivars (the early-ripening cv. Zahir and the late-ripening cv. Ruta). Two experimental seasons: 7 May 2017 to 9 July 2017 and 19 November 2017 to 10 April 2018 were monitored in a greenhouse. In the first period, air temperature exceeded 40 °C during growth in the summer months and hence shortened the growing season. The layout was based on lysimeters distributed in random split-plots blocks. The main variable (Table 1) was N-fertilization level, with four N levels applied via drip irrigation. The secondary variable was the wheat cultivar.

**Table 1.** **N** treatments for wheat grown in the greenhouse.

| Treatment No. | N Application |
|---|---|
| 1 | No N application |
| 2 | 60 Kg N ha$^{-1}$: basal application only |
| 3 | 120 Kg N ha$^{-1}$: 60 basal + 60 topdressing |
| 4 | 180 Kg N ha$^{-1}$: 60 basal + 120 topdressing |

Note: 10 Kg N ha$^{-1}$ = 1 unit of total N.

Each replicate was grown in a 10-liter lysimeter that contained perlite as growing substrate. In addition, 12 ppm phosphorus (P) and 50 ppm potassium (K) were applied during growth period in all treatments.

The final stand consisted of approximately 300 plants m$^{-2}$. Plants were photographed with a smartphone camera at the four stages, above. Samples were immediately taken for the laboratory analysis. For cv. Zahir, photographs and samples were taken at 29, 42, 50 and 65 DAE, and for cv. Ruta at 29, 42, 58 and 79 DAE. We have to consider that growth is enhanced in the greenhouse compared to the field. The unique approach in this study was the use of a RGB photograph to estimate %N. The last were calibrated with the laboratory N analysis. For the calculation of total N uptake, DM was weighed and the value multiplied by %N/100.

Details of the processes which involve computerized machine learning and artificial neural network are given in Zilberman et al. [4,5].

### 2.3. Field Experiments

Wheat was sown on 20 Nov 2018 and complete emergence was 25 days later. The wheat was harvested on 21 May 2019. Rain from October to May was 350 mm and average temperature in December 2018 was 15 °C and in May 2019, 22 °C. The soil was classified as loess, with the equivalent USDA classification of Torritfluvent. It consisted of about 17–21% clay, 15–23% silt, and the rest was sand. The experimental field 2000-m$^2$ was located in Saad, Israel (31°28′04.44″ N, 34°32′16.70″ E). The experimental design was random split-plots in five blocks. The main variable was N treatment, with six levels (Table 2), and the secondary variable was wheat cultivar. We used the early-ripening cv. Amit, late-ripening cv. Ruta and intermediate-ripening cv. Kitain. In total, there were 72 plots, each one 10 m in length and 6m width. N topdressing for treatments 4, 5 and 6 (Table 2) was applied at the 6- to 7-leaf stage (GS 14 according to Zadoks [18].

**Table 2.** Nitrogen application treatments (kg pure N ha$^{-1}$) in the field experiment.

| Treatment No. | N Application |
|---|---|
| 1 | 40 Kg N ha$^{-1}$: basal application only |
| 2 | 100 Kg N ha$^{-1}$: basal application only |
| 3 | 150 Kg N ha$^{-1}$: basal application only |
| 4 | 90 Kg N ha$^{-1}$: 40 basal + 50 topdressing |
| 5 | 150 Kg N ha$^{-1}$: 100 basal + 50 topdressing |
| 6 | 200 Kg N ha$^{-1}$: 150 basal + 50 topdressing |

### 2.4. Biomass Sampling (Greenhouse and Field Experiments)

In the greenhouse experiment, 5 plants were sampled from each lysimeter; in the field experiment, in each replication, all plants in a 0.5-m$^2$ area were sampled. Biomass was determined at the growth stages as indicated above.

A representative sample was cut, washed and oven-dried at 65–70 °C, before its dry weight determined. The sample was then milled and analyzed in the laboratory to estimate

the total N content with a rapid N exceed device (Elementar Germany). Total areal N content was calculated as DM $\times$ %N/100.

### 2.5. Input/Output Data

In this technology the only sensor is the plant and the data logger is the smartphone. Thus, the input includes planting or sowing day, crop stands (number of plants per square meter) and the formula of the fertilizer. The photograph of the plant or a group of plants was taken and sent to the application to develop %N and nitrogen uptake. The rest of the input data is stored in the algorithm. It contains continuous estimates of a day by day DM during the growing season, and the calibration equation. The N uptake output is a product of two components %N and DM from emergence to sampling date. To help potential users to save time and labor for biomass sampling we selected widely used Decision Support System for Agricultural Technology (DSSAT)crop simulation model (Hogeboom et al.; [19]). with a daily time step from sowing to maturity.

DSSAT was calibrated against experimental data under growth conditions according to Bar Yosef and Ben Asher [20]. Simulation with two sets of data: Soil and Weather files were used for canopy and grain yield tests. The best fit regression equation was: DSSAT results = $1.1 \times$ experimental results $-824$; $r^2 = 0.91$.

Based on the successful calibration of the model a data table of daily canopy weight was stored in the algorithm and used to calculate the suitable Nfuptake for every sampling date that provided N% from the RGB photographs.

Using DSSAT optimal nitrogen uptake as a reference, two output options were considered.

Measured nitrogen uptake = N% $\times$ DM/100 < DSSAT simulation optimal uptake, the application displays "add X Kg of N per Ha" and according to the selected fertilizer, how much fertilizer should be applied per hectare.

Measured nitrogen uptake $\geq$ DSSAT simulation of optimal uptake the screen shows "no need to fertilize".

### 2.6. Statistical Analysis

The analysis was divided to two parts: descriptive statistics and accuracy measurements of RGB output compared to laboratory results.

The descriptive statistic included average and standard deviation (SD) of all comparisons between laboratory references and RGB outputs. It included also the mathematical relationship between laboratory reference and the RGB output by a typical correlation analysis. It is considered best approximation of all the individual measurements (lab versus RGB outputs.). The strength of the relationship between the two variables was measured by the correlation coefficient known as $r^2$. However we observed a distinct increasing variance, which renders ordinary least squares inadequate to account for the relationship between N uptake measured in the laboratory and N uptake sensed remotely. We therefore refitted the model using non linear regression analysis to find the best unbiased linear predictor. However, the difference between the linear and the non linear model was too small and we preferred to use a confidence band as it appears in Figure 1. The correlation between laboratory and camera analyses was plotted by Sigma plot using two software linear regression and nonlinear two steps regression.

The accuracy of the RGB output was estimated by three methods:

(A) Root mean square error (RMSE) to evaluate usefulness and accuracy of the RGB trained by ANN software.

$$\text{Normalized RMSE (lab vs Cam)} = (\sqrt{(\sum_{i=1}^{n} \frac{(labi - cami)^2}{n})})/labi \qquad (2)$$

where labi is the ith measurement of laboratory Nuptake analysis and cami is the ith result of the trained RGB photo. The number of comparisons made during the three experiments is n $\approx$ 90.

RMSE was solved by Excel as in Equation (2)
RMSE = SQRT(SUMSQ($C_1$:$C_n$)/COUNTA($C_1$:$C_n$)).

(A)　Relative accuracy/error. It refers to the closeness of a measured value to a "true" (reference) value. The "True" Nuptake value corresponds to the uptake of N measured by standard chemical laboratory.

$$\text{Relarive Error}\% = \left| \frac{CAM\ Nuptake - LAB\ Nuptake}{LAB\ Nuptake} \right| = \frac{Absolute\ error}{\text{"True" } value} \times 100 \quad (3)$$

$$\text{Accuracy} = 1 - \text{Relative error} \quad (4)$$

Equation (4) gives the percent to which result of the camera RGB are conformed to the correct/reference values that are the laboratory analyses.

(A)　Data significance measure: Excel two tails at $p = 0.05$ $t$-test option was used to evaluate the significance level of the results by comparisons between laboratory and RGB results.

Sigma plot was also used to calculate and plot the parameters of the Nc model with the confidence bands for the data. We assume that the soil parameters were uniform and therefore, average values were only affected by the main variable (fertilizer application) and secondary variable (cultivar).

### 2.7. Summarizing Flowchart

The processing of the photos taken by a smartphone is summarized by a flowchart that is given below (Figure 2). The scheme describes a process that stats when a photo of the wheat is taken and immediately sent from the camera through the internet cloud to a server. The server calculates nitrogen uptake from DAE to the date of photographing and amount of recommended fertilization for optimal production.

### 2.8. Theory:The Algorithm for Fertilization Decision

The following sections preset a set of equations that are based on %N which is measured by RGB photographs taken with smartphones.

### 2.8.1. Determination of Nc: New Approach

Nitrogen dilution curves relate a crop's critical nitrogen concentration (%Nc) to biomass (W) according to the allometric model Equation (1): %Nc = a $W^{-b}$. The theoretical foundations of this model are strong [10,12] and therefore it gained the absolute majority of users. The dilution of nitrogen occurs because at the start of the growing season, biomass mostly consists of leaves with high proportion of metabolic tissue and high nitrogen concentration, but as the crop grows, relatively more structural tissue, i.e., stem with a smaller nitrogen concentration, is produced. This equation describes the onto-genetic decline of average %N as a function of canopy DM which in itself is increasing with DAE. Therefore, Equation (1) for determination of %Nc is indirectly connected with the physiological time cycle of N uptake at canopy level. However, despite its high agronomic relevance, Nc cannot be used directly in farm conditions because it has several disadvantages; first and most important is that DM weight determination is very time consuming. Secondly its initial value at DM = 0 is not known. Thirdly its value near zero is unrealistic because it goes to infinity and fourth it only provides %N and not N uptake (Kg ha$^{-1}$) which is an important parameter for N management. So, it is imperative to develop indirect methods for Nc estimation through more operational procedures. Very few methods have been proposed for indirect method that are suitable to adopt smartphone as a diagnostic tool. Some of them are specified in Section 4.3 but the calibration or validation of these methods with Nc have not been always made and, when they have been, they did not give univocal relationships. Two other relevant indirect methods were not discussed in Section 4.3 and their comparative advantage is that they are based on days after sowing.

One of them was based on leaf area duration (LAD) [21] and the other on thermal days [22]. General expression of the time dependent curves of Equation (1) is given by: %Nc = a $X^{-b}$ where X can be one of the three variables: 1. W, the classical biomass model 2. Leaf area duration LAD=$\int_0^t LAI \, dt$ (Days) that is based on LAI (the leaf area index) and 3. the thermal time i.e., TTD=$\int_0^t \overline{(T-T0)} \, dt$ that is based on average daily temperature. Where TTD is thermal time duration (_ºC d or degree-days). T is the daily average temperature and T0 is a reference temperature. The second method allows the estimation of actual crop mass, through LAI. The third method emphasizes the environmental growth conditions. When %Nc wasplotted against thermal time from sowing the $r^2$ of 0.82 for the thermal time, the model was slightly better than that for the biomass (W) model. Of course it was more operational than the original equation because today temperatures can be stored on local computing servers. The two modifications of Equation (1) are more practical than the original equation but are still consuming measuring time and near zero they are not realistic.

An improvement of allometric model Equation (1) while following the fundamental strength of the original theory is given in Equation (5).

$$Nc_{(t)} = Nc_{(t=0)} / [1 + (DAE/DAE_{50})^{-b}] \tag{5}$$

Equation (5) is an inverse logistic function that based on days after emergence.

Where $Nc_{(t=0)}$ is the Nc when DAE = 0. and $DAE_{50}$ is the day when $Nc_{(t)} = 0.5* Nc_{(t=0)}$.

Similar to Wang et al. [21] and Hoogmoed and Sadras [22], the %N contents versus DM was replaced by time indices such as LAD, TTD or DAE.

For practical purposes, the simplest DAE parameter can be used as a surrogate to DM measurement and saves time and labor.

In Equation (5) $DAE/DAE_{50}$ ratio is relative time (dimensionless) and b is a parameter indicating N-dilution rate: the larger b is, the faster the rate of N dilution. Further examination of Equation (5) suggests that Nc = 0.5 × $Nc_{(t=0)}$ and then DAE = $DAE_{50}$. Thus, Equation (5) resolves the singularity problem in the general expression of Equation (1) at which near zero DM, LAD or TTD, concentration of nitrogen in leaf are infinitely large. On the other hand close to the emergence days [≈zero in Equation (5)] before intensive nitrogen dilution %N is at its maximum and %N/2 is at $DAE/DAE_{50}$ = 1 as shown in Figure 5.

To resolve the singularity problem that expose infinite %N when DM ≈ 0 in Equation (1) it is legitimate to take N content in the soil plus fertilizer application as Nc at zero [21,22]. The values are known before planting or sowing (DAP = 0) and are included as a data point in Equation (5).

### 2.8.2. Determination of NNI

Nitrogen Nutritional Index, NNI, describes the ratio between the measured or actual %N at time "t" and the critical or potential N concentration ($Nc_{(t)}$) at the same time ($N_{(t)}/Nc_{(t)}$); where "t" is DAE [19]. A general way to obtain NNI experimentally is to plot $\%N_{(t)}$ vs. $\%Nc_{(t)}$. Apparently, there is a linear relationship between $N_{(t)}$ and $Nc_{(t)}$, and the slope of this plot expresses NNI. It can also be calculated from the weight of the Nuptake that has been taken up [NNI = N (Kg ha$^{-1}$)/Nc × (Kg ha$^{-1}$)].

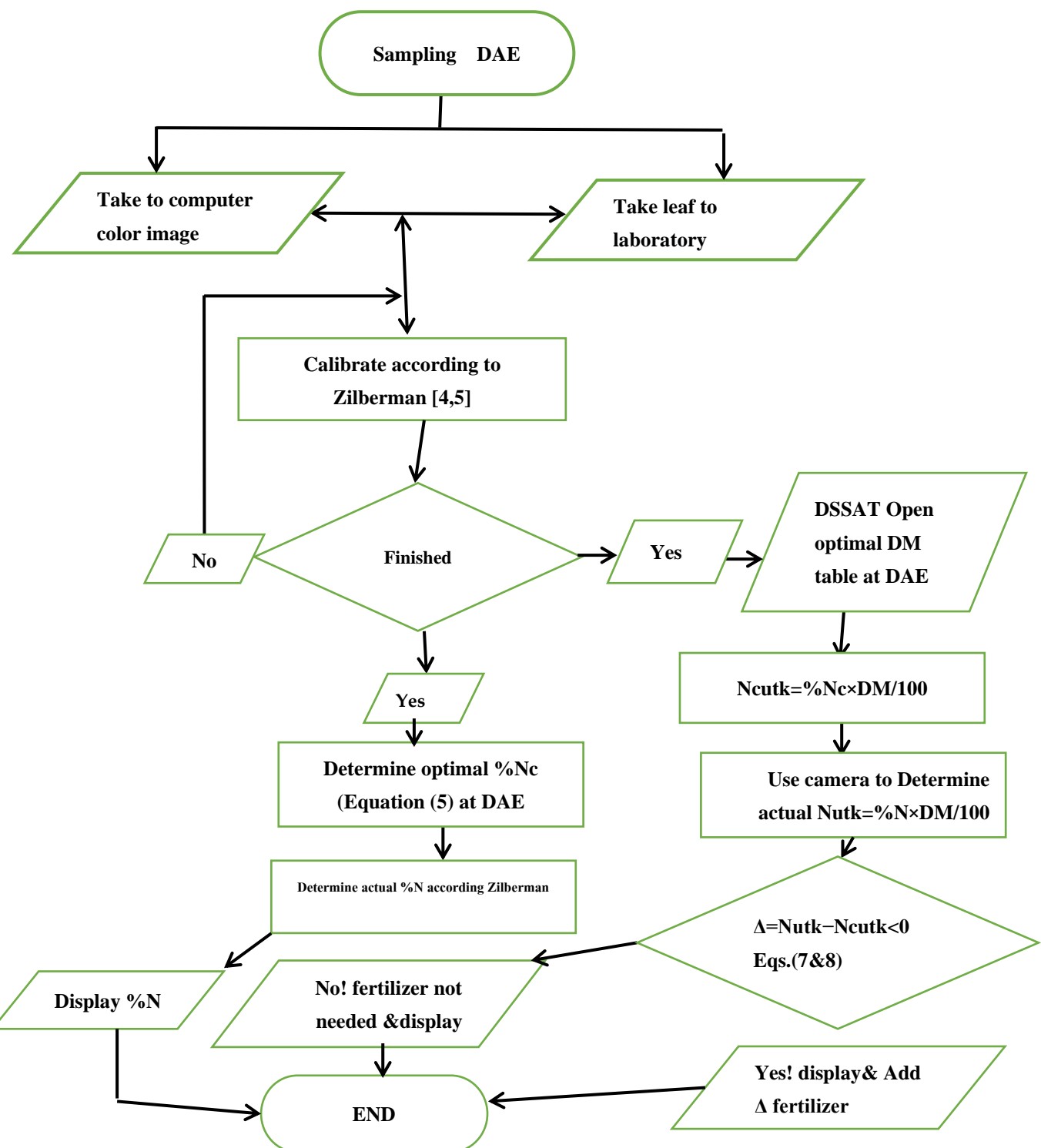

**Figure 2.** Flowchart describing the processing of photos taken by a smartphone (UTK = uptake ).

Specifically, using Equation (5) for critical N of wheat, NNI is extended to Equation (6)

$$NNI_{(t)} = DM_{(t)}N_{(t)} / \{DMc_{(t)}Nc_{(t)} / [1 + (DAE/DAE_{50})^{-b}]\} \tag{6}$$

where DMc (Kg ha$^{-1}$) is the dry matter of a field in which %N is at its best conditions, %Nc.

For efficient nutritional management of wheat, information on its total N status within fields is of utmost importance. The grower needs this information from 3–5 leaves stage onward in order to achieve appropriate fertilization dose and timing. Currently

management is mainly based on %N and qualitative expert knowledge but it is preferred to have quantitative information on the actual status at the right moment. In spite of the fact that destructive DM measurements have been very tedious, time-consuming and not practically applicable, actually, some authors [19,20] have used them only to calibrate models but due to the difficulties they were quite limited on commercial farmers' fields.

### 2.8.3. Determining N Deficiency

The difference between optimal nutrition and insufficient N application (Δ) can be used to determine N deficiency, which needs to be known to obtain optimal yield. The weight of the N (Kg ha$^{-1}$) taken up by a crop enables the grower to obtain optimal yield as:

$$\Delta = \frac{Nc(t) - N(t)}{Nc(t)} = 1 - \frac{N(t)}{Nc(t)} \tag{7}$$

Δ is the difference between %Nc and %N at time "t". It is needed for calculation of application dose of N until Nc is obtained. The value of Δ ranges between 0 and 1 ($0 \leq \Delta \leq 1$). At Δ = 0, fertilizer is not needed and when $\Delta \leq 1$, N should be added (see it also in the flowchart above). The amount of N deficiency (Kg ha$^{-2}$) is given by:

$$\text{N deficiency} = [(DMc_{(t)} \times Nc_{(t)}) \times (1—NNI)] \times 10{,}000 \text{ m}^2 \text{ ha}^{-1} \tag{8}$$

where: N deficiency is the amount of N that should be added to one hectare (Kg ha$^{-1}$).

## 3. Results and Discussion

### 3.1. Comparison of Digital Camera and Laboratory Test Results

The main objective of this study was to relate N uptake measured in the reference laboratory to that obtained with a simpler diagnostic tool (RGB camera) that monitored in three wheat experiments and about 90 laboratory tests.

The correlation between nitrogen uptake measured by smartphone camera and laboratory analysis is displayed in Table 3 and Figure 3.

**Table 3.** Correlation between N uptake determined by the camera method and in the laboratory results from greenhouse (2018) and field (2019) experiments.

| No. of Leaves | Cultivar | Year | Slope * | $r^2$ |
|---|---|---|---|---|
| 3–4 | Zahir | 2018 | 0.78 | 0.84 |
| 3–4 | Ruta | 2018 | 0.92 | 0.88 |
| 6–7 | Zahir | 2018 | 0.98 | 0.91 |
| 6–7 | Ruta | 2018 | 1.07 | 0.93 |
| Flag leaf | Zahir | 2018 | 0.95 | 0.93 |
| Flag leaf | Ruta | 2018 | 1.02 | 0.92 |
| 3–4 | 3 cultivars | 2019 | 1.00 | 0.85 |
| 6–7 | 3 cultivars | 2019 | 1.00 | 0.82 |
| Heading | 2 cultivars | 2019 | 1.00 | 0.72 |
| Heading | Ruta | 2019 | 1.05 | 0.77 |
| Average | All cultivars | 2018–2019 | 0.98 | 0.86 |
| SD | All cultivars | 2018–2019 | 0.08 | 0.07 |

* N uptake by Camera = slope × N uptake by laboratory.

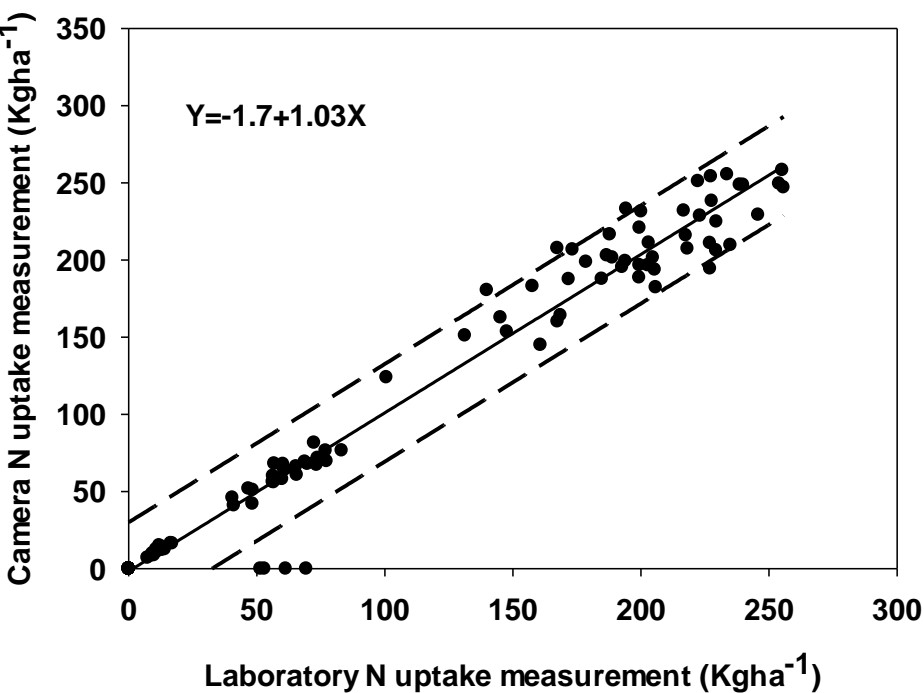

**Figure 3.** N uptake as determined by the camera (cam) and in the laboratory (lab) ($\approx$90 points). The correlation includes all wheat cultivars and the three experiments. Dashed lines are the boundaries of the 95% confidence band.

An average slope of about one unit (0.98 $\pm$ 0.08; $r^2$ = 0.86) is acceptable but there are deviations that slightly reduced the goodness of slope at 3–4 leaf stage and $r^2$ at heading stage in Table 3. The reduced value of the slope at 3–4 leaf stage resulted from small number and size of calibrated leaves relative to soil surface [23] that occupied the field of view of the camera. At heading stage in the field experiment, $r^2$ was relatively low because the greenness of the leaf was masked by pale green color of the spikes. It reduced the strength of the relationship.

The average N uptake $\pm$ SD was 250.93 $\pm$ 100.6 and 259.91 $\pm$ 110 Kg ha$^{-1}$ for laboratory and camera respectively and the correlation between them was very good through the entire growing stages Figure 3 and Equation (9).

$$[\text{camera mesuration} = -1.7 + 1.03 \times (\text{lab mearement}); r^2 = 0.98] \qquad (9)$$

In addition we used RMSE Equation (2) in its normalized form to measure the concentration of the data around the line of best fit. This statistical analysis is displayed in Figure 2.

As was mentioned in the perviou Section 2.6 the increasing variance in Figure 3 may render ordinary least squares inadequate to account for the relationship between N uptake measured in the laboratory and N uptake sensed remotely. It was therefore important for this analysis to include the confidence limit band (Dashed line) to find the best unbiased linear predictor. It can be seen that the deviation from the 1:1 line was very small and the increasing variance did not affect the general agreement between the measurement of N uptake by camera or by laboratory.

Figures 3 and 4A,B are corroborating evidence that smartphones RGB photographs' can be useful because they correlated well with laboratory tests (Figure 3), include 80% of relative accuracy and low RMSE < 0.2 (Figure 4A,B). Because there is no upper limit on a percent accuracy or lower limit of RMSE the boundaries are only subjective judgment on whether the data is useful or not.

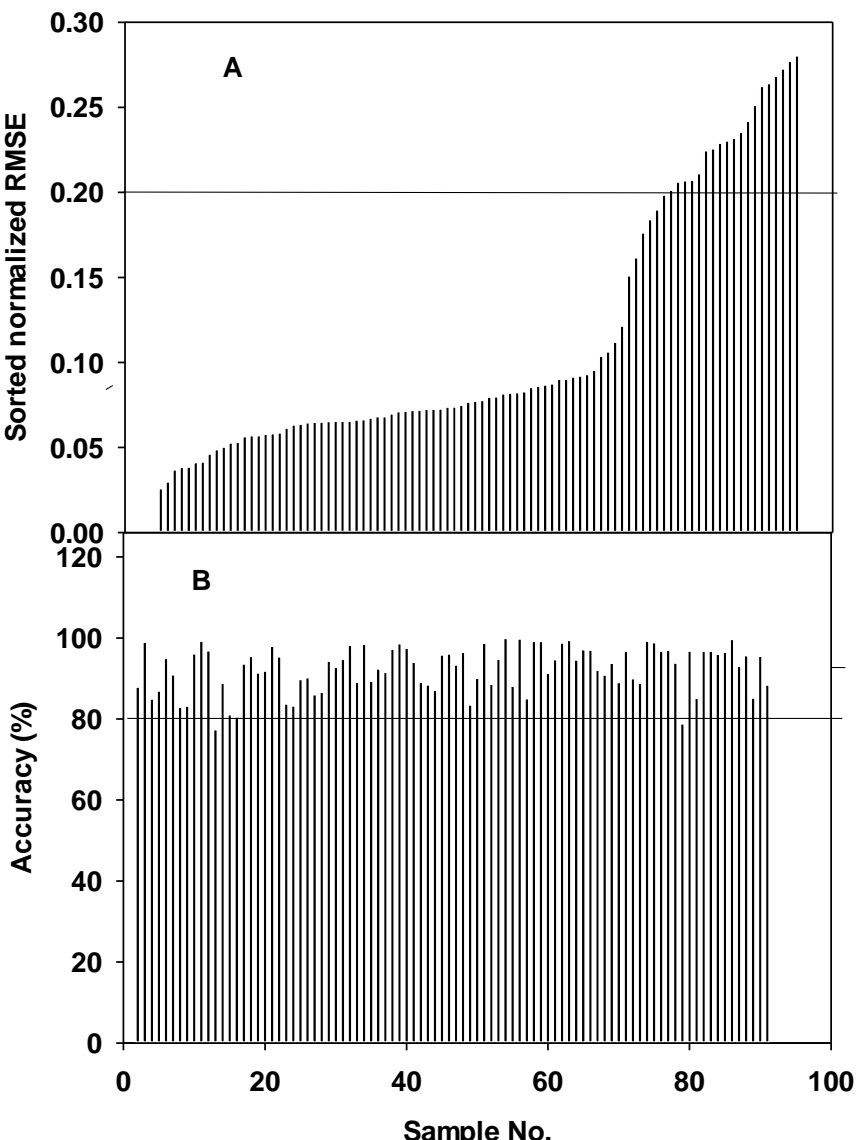

**Figure 4.** Statistical comparison between standard laboratories tests and photographic measurements of %N. (**A**) Normalized root mean square error (RMSE) of %N. Equation (2) The horizontal solid line is a separator between errors above and below 0.2 (**B**). Percent accuracy of N% Equation (4). The horizontal solid line is a separator between smartphone accuracy of N% above and below 80% agreements with laboratory reference.

Bearing in mind the other factors that affect greenness of the photographs and N fertilization decision the technology is taking into account most of the factors. For example soil pixels are eliminated from the image analysis by preprocessing and computerized features extraction. Others such as nitrogen leaching beyond the root zone by rainfall and nonuniform N fertilizer distribution is expressed by the photographic output. Among others spatial variability of nitrogen concentration on the field is using the crop as a sensor to describe it and the smartphone as a data logger to calculate it.

*3.2. Experimental Study of Nc, NNI and Their Combination to Support Fertilization Decisions*
3.2.1. Determination of Nc for Wheat

The dilution rate of Nc as the growing seasons progresses is displayed by inverted logistic curve in Figure 5.

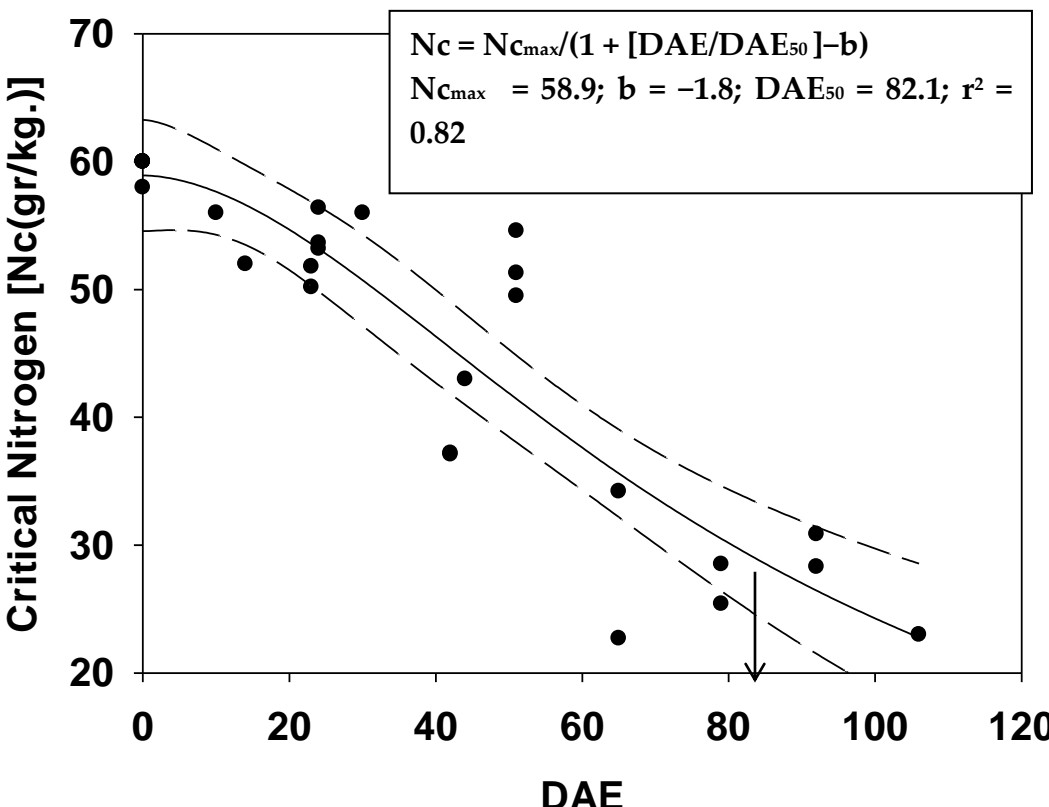

**Figure 5.** Dilution curve of critical nitrogen (Nc) vs. days after emergence (DAE).

The points are the highest experimental values of Nc found during the sampling days. The arrow indicates the value of $DAE_{50}$, obtained when $Nc = Nc_{max}/2$, $\approx 30$. The band between the dashed lines is the 95% confidence limit.

The arrow points to 82 DAE as $DAE_{50}$ when in Equation (5), Nc is diluted to about 30 g kg$^{-1}$ or half (50%) of its initial (maximum) concentration. Note that the abscissa in Figure 5 shows DAE instead of canopy yield (DM) and best fit parameters of Equation (5) for wheat are displayed on top of it. Most studies, preferred DM yield units on the abscissa but it requires time and manpower to obtain. Moreover, Nc in Equation (1) tends to unrealistic infinity when DM << 1 Mg ha$^{-1}$. On the other hand, the best fit line of the data in Figure 5, which is based on Equation (5) provides a realistic general model. At DAE $\approx 0$, C is at its maximum because uptake from soil and applied fertilizer is only at its initial stage. That is, $Nc_{(t=0)} \approx Nc_{max}$ and it can be a priory known as initial Nc. Moreover, by definition Nc $_{(t=DAE50)} = 0.5 \times Nc_{max}$, leaving "b" as the only unknown in Equation (5) [24].

In Figure 5 the dilution curve is described properly by inverted logistic approach. It, however, does not consider environmental conditions that could affect plant growth. For instance, differences in temperature and water availability would have considerable impact on the amount of biomass produced. This may affect the dilution rate of critical N concentration which is determined by "b" in Equation (5).

### 3.2.2. Determination of Nitrogen Nutrition Index (NNI)

Nitrogen nutrition index (NNI) is a diagnostic tool that is calculated as the ratio between the measured and the critical N content in Equation (6). It indicates the minimum N content required for the maximum biomass production [25].

An example of values calculated from Equation (8) is displayed in Table 4 for two wheat cultivars (Zahir and Ruta).

**Table 4.** Correlations between $N_{actual}$ and Nc in the greenhouse experiments. The slope of the best-fitting linear line is the ratio between actual and potential uptake of N which, by definition, is NNI.

| N Application (kg 10,000 m$^{-2}$) * | Correlation Equation Based on N Uptake (kg 10,000 m$^{-2}$) | r$^2$ |
|---|---|---|
| 0 | $0.2Nc = N_{actual}$ | 0.63 |
| 60 | $0.34\ Nc = N_{actual}$ | 0.83 |
| 120 | $0.83Nc = N_{actual}$ | 0.83 |
| 180 | $Nc = N_{actual}$ ** | 1 |

* 1 unit of N application = 10 kg 10,000 m$^{-2}$. ** Nc = 18 N units.

The greenhouse experiments (Figure 6), provided significant differences between the slopes describing NNI as the ratio between N and Nc uptake. The numerical values of Figure 6 are summarized in Table 4, from which two examples (slopes of 0.2 and 0.8) were taken to display calculated N deficiencies in Table 4.

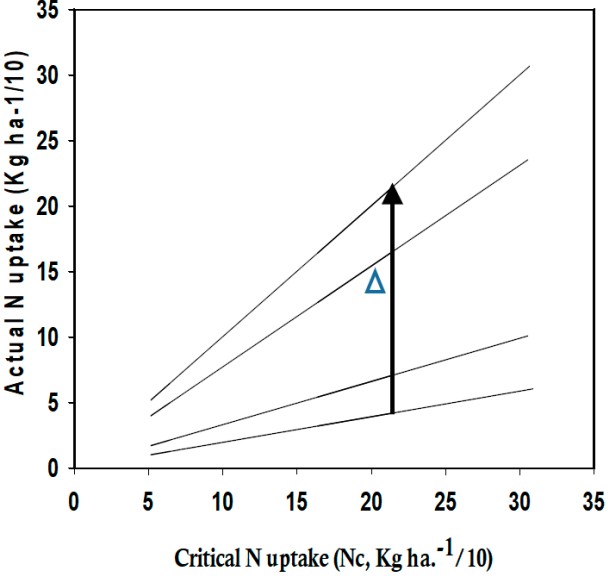

**Figure 6.** Correlation between limited applications of N (0, 6, 12 N units) and 18 critical N (Nc). The slope (uptake of actual N vs. uptake of Nc) is the Nitrogen Nutritional Index (NNI) for each N level.

Results of the correlation analysis are given in Table 4. The arrow indicates the difference ($\Delta$) between the lowest N application and Nc.

In Figure 6, the lower line (0) represents the NNI of a treatment without N application. In this treatment, N content represents only the N in the substrate. The two middle treatments (6 and 12) represent deficient N application. The upper solid line (18) represents the course of Nc uptake. An example of deficient N uptake that should be increased to reach the Nc level is marked with an arrow and $\Delta$ Equation (7).

The upper line in Figure 6 has a slope of 1, and the coefficient of determination (r$^2$) is also 1 because the two axes are based on the same Nc value. Note that NNI determined by the slopes in Figure 6 responded properly to the experimental design. That is, NNI was higher when N application was larger. In Table 4, the slopes (0.2, 0.34, and 0.83) were calculated from actual data points together with the correlation coefficient (r$^2$), which was acceptable for the designed experiments.

### 3.2.3. Combining NNI and Nc to Determine N Deficiency

Determining %N with a simple RGB camera, offers a straightforward, rapid and inexpensive method for the management of N application. This remote sensing integrated

approach combines NNI and Nc measurements to determine N deficiency with acceptable accuracy as demonstrated in Figures 1 and 2.

There are two ways to calculate NNI, one on the basis of %N and the other on the basis of N uptake. The former saves on the labor involved in collecting crop biomass. On the other hand, NNI calculated on the basis of N uptake is more reliable. Nevertheless, this study indicated that despite the differences in reliability between the two approaches, the proposed photographic method about the N status of the wheat stands by using proximal reflectance measurements can be useful to support precis N fertilizer applications.

The data in Table 5, were calculated using Equation (10):

$$\Delta = Nc - N_{actual} = Nc - Nc \times NNI = Nc\,(1-NNI) \tag{10}$$

where $\Delta$ = N deficiency or N units (Kg ha$^{-1}$) that should be added,

$N_C$ = %Nc $\times$ DMc/100 Kg ha$^{-1}$

NNI = $N_{actual(t)}/Nc_{(t)}$ and $N_{actual(t)}$ is the actual N uptake (Kg ha$^{-1}$).

**Table 5.** Nitrogen deficiency (Kg ha$^{-1}$) calculated for two values of NNI (0.2 and 0.8).

| DAE | DMc | STD | Fractional Nc | NNI = Nuptake/Nc | N Deficiency | NNI = Nuptake/Nc | N Deficiency |
|---|---|---|---|---|---|---|---|
| | (Kg ha$^{-1}$) | (Kg ha$^{-1}$) | (%/100) | | (Kg ha$^{-1}$) | | (Kg ha$^{-1}$) |
| 0 | 0 | | 0.06 | 0.2 | 0 | 0.8 | 0 |
| 23 | 352 | 48 | 0.05 | 0.2 | 14.1 | 0.8 | 3.5 |
| 42 | 1632 | 245 | 0.04 | 0.2 | 52.2 | 0.8 | 13.1 |
| 65 | 2848 | 256 | 0.03 | 0.2 | 68.4 | 0.8 | 17.1 |
| 0 | 0 | | 0.06 | 0.2 | 0 | 0.8 | 0 |
| 23 | 448 | 40 | 0.05 | 0.2 | 17.9 | 0.8 | 4.5 |
| 42 | 1504 | 180 | 0.04 | 0.2 | 48.1 | 0.8 | 12 |
| 65 | 3744 | 562 | 0.03 | 0.2 | 89.9 | 0.8 | 22.5 |
| 79 | 11584 | 2317 | 0.03 | 0.2 | 278 | 0.8 | 69.5 |
| 0 | 0 | | 0.06 | 0.2 | 0 | 0.8 | 0 |
| 10 | 80 | 9 | 0.06 | 0.2 | 3.8 | 0.8 | 1 |
| 14 | 158 | 22 | 0.05 | 0.2 | 6.3 | 0.8 | 1.6 |
| 30 | 1818 | 145 | 0.06 | 0.2 | 87.3 | 0.8 | 21.8 |
| 44 | 4064 | 57 | 0.04 | 0.2 | 130 | 0.8 | 32.5 |

The linearity of the NNI model shown in Table 4 and Figure 5 was maintained by restricting the sampling dates to the flag leaf stage (50 and 58 DAE). The N deficiency values in Table 5 at main growth stages in which plants are still responding to fertilization can therefore be used to assess the nutritional status of spring wheat. Based on the good agreement between laboratory and RGB photographs, results in Table 5 show that NNI and fertilization dose can be obtained fast and timely by proximal sensing with a smartphone.

3.2.4. Accumulation of DM in the Lysimeters Experiments and Nitrogen use Efficiency (NUE)

Proximal sensing of Nitrogen management and use of combined NNI and Nc can improve N efficiency and reduce residual N in the soil without reducing canopy or grain yield. Therefore as the highest ratio: canopy weight /applied Nitrogen is of utmost importance. The data specified in Figure 7 was used to calculate the canopy NUE.

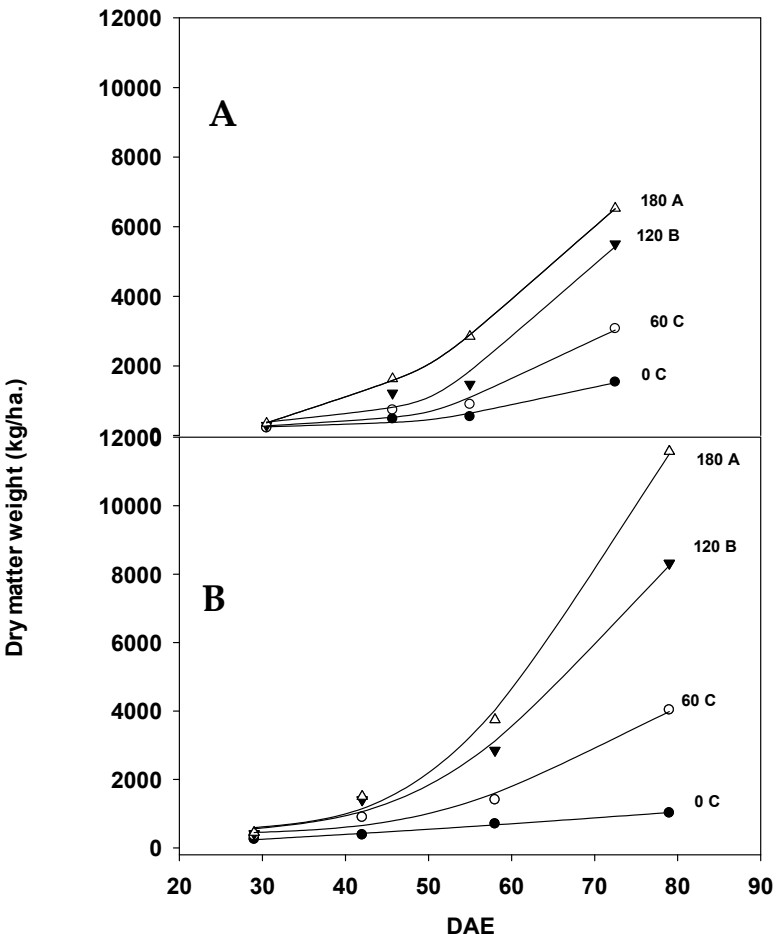

**Figure 7.** Accumulation of dry matter vs. days after emergence (DAE) for (**A**) early-ripening cv. Zahir and (**B**) late-ripening cv. Ruta during growth. Different uppercase letters indicate significant differences at $p < 0.01$. The numbers inside the graph indicate applied pure N in Kg ha$^{-1}$, equivalent to 0–18 N units.

There were distinct differences in DM weight between the cultivars and treatments. The early-ripening cultivar that received only 6 N units reached a maximum 3100 Kg DM ha$^{-1}$ and that receiving 18 N units reached a maximum 6800 Kg DM ha$^{-1}$ at heading. Heading of the late-ripening cultivar occurred 79 DAE or two weeks after heading of the early-ripening variety. DM weight at heading was about 4000 Kg ha$^{-1}$ for 6 N units and 11,600 Kg ha$^{-1}$ for 18 N units. It should be noticed that there were no significant differences between varieties during the first 40–50 DAE.

A comparable NUE with canopy yield can be seen in Table 6.

**Table 6.** The effect of Nitrogen application dose on final canopy weight and Nitrogen use efficiency (NUE).

| Treatment No. | 2± | 2 SDEV | 3± | 3 SDEV | 4± | 4 SDEV |
|---|---|---|---|---|---|---|
| Total N units Kg ha$^{-1}$ | 60 | 60 | 120 | 120 | 180 | 180 |
| DM Zahir Kg ha$^{-1}$ | 3100 | 310 | 6192 | 742 | 6800 | 884 |
| NUE Kg DM Kg N$^{-1}$ | 52 | 4.7 | 52 | 5.2 | 38 | 4.2 |
| DM Ruta Kg ha$^{-1}$ | 4000 | 480 | 9360 | 842 | 11600 | 1160 |
| NUE Kg DM Kg N$^{-1}$ | 67 | 10 | 78 | 6.24 | 64 | 5.8 |

In Table 6 the two varieties reached peak NUE at 120 Kg N/ha. and the lowest at 180 Kg N/ha. That is high canopy yield does not mean high NUE and according to the

table above and Figure 6 early ripening variety has lower canopy yield and lower NUE.than late ripening variety.

## 4. Discussion

### 4.1. Proximal Nitrogen Sensing

Proximal nitrogen sensing by a still camera can help farmers to apply the right nitrogen input, in the right amount at the right time. It is based on a set of equations that successfully calibrated against laboratory analyses. Positive relationship, based on data from all field samplings at various dates and wheat varieties, was expressed by a significant linear function of RGB camera photo vs laboratory output. Such a relationship between camera and laboratory data (Figure 3) has also been reported [4]. with other crops indicating the generality of the approach.

Literature to date is short of papers integrating proximal measuring method and modified known equations quantifying wheat canopy production. Therefore, our work fills this gap. It offers applicable contribution to the present literature but with practical adjustments of the methodology.

In addition, modification of critical nitrogen (Nc) equation and its parameters for spring wheat is presented. In order to reduce time and labor required to harvest dry and weigh to obtain DM for Equation (1), we proposed an inverse logistic function Equation (5) to describe Nc as a function of DAE. A basic assumption behind the replacement of DM by DAE is that DM is a linear function of DAE. However, from Figure 7 it was clear that the last is not linear but a second order polynomial equation with $r^2 \approx 1$ that is changing with time wheat variety and treatments. Parallel, testing linearity approximation revealed that a linear curve fit of DM vs DAE yielded that $r^2$ varying from 0.86 to 0.98. which is acceptable approximation for field experiments. The validity of Equation (1) was tested for winter [14] and spring wheat [26] for the effect of field situation and concluded that in spite of large differences in growth rate, cultivar, soil and climatic conditions shoot biomass it can be said that 'critical nitrogen dilution curve is unique.

Currently, the model is an integrated algorithm that include %N from RGB photograph, Nc and NNI and ΔNNI from Equations (5)–(7) and DSSAT [19] output of daily top weight for wheat.

DSSAT is available model and in the algorithm we used it to determine a target function of dry matter accumulation. In the algorithm DM at any DAE is multiplied by %N to obtain Nuptake at a given phenological stage. The output is used to support fertilization decision. We presented here a direct measuring method in which the plant itself is the sensor and the smartphone is used as data logger. The advantages of smartphone include availability simplicity, real-time flow of data, and inexpensiveness.

### 4.2. Other Proximal Sensing Methods

Aranguren et al. [23] reviewed large number of publications that dealt with proximal sensing including.

Yara N-TesterTM and RapidScan CS-45, diagnosed the N nutritional status of wheat.

They adjusted the handheld instruments according to the NNI, and obtained indications for the magnitude of the N surplus or deficiency. In their experiments with topdressings, all treatments were under N deficiency (NNI < 0.9), even the overfertilized ones.

However, the NNI was able to detect differences among treatments.

Similarly in our study we detected differences among treatments (see Table 5) but due to the determination of $Nc_t$, (where NNI = 1) at several growing stages, it was possible to obtain calculated indication of the magnitude of recommended nitrogen topdressing fertilization at various growing stages.

Another aspect that worth comparison between our study and theirs [27] is the NDVI and NDRE (Normalized Difference Red Edge). Some of the authors quoted by Aranguren et al. [23] highlighted that, when the wheat canopy is not closed, soil background exposure reduces the reliability of using reflectance for the estimation of crop N status. We

distinguished similar phenomena in our study but were able to solve it by a photographic closeup with the smartphone while the NDVI that is taken from satellites or by heavy instrumentation is not as flexible.

Spectral data collected by RapidScan CS-45 were converted into NDVI measurements or NDRE by calculating vegetation indices. However, at the end of the growing cycle, the use of NDVI is suitable but less accurate than the handheld devices such as RapidScan CS-45 which is a portable entirely self-contained ground-based active crop canopy sensor that integrates a data logger, graphical display GPS and more. However comparing to cellular phone they are much more expensive and its availability is limited but It measures crop reflectance at 670, 730 and 780 nm and provides the NDVI and NDRE and therefore is versatile.

### 4.3. Indirect Methods

To make fertilizer recommendation, the alternative widespread method consists in plant analysis operating manually with handheld instruments or send to laboratories.

Handheld devices: there are methods for measuring the nitrate content of the sap at the base of the petioles (Rosen et al. [28]) and those giving relative measurements of leaf N content by using different hand-held chlorophyll meters, such as SPAD-502 (Konica Minolta, Tokyo, Japan) and Hydro N Tester (Yara International ASA, Oslo, Norway;). Our proposed method is based on RGB reflected radiation. Reflection by vegetation in the visible region (RGB) of the electromagnetic spectrum is predominantly influenced by chlorophyll pigments in the leaf tissues. The principle of SPAD on the other hand is based on components of radiation balance, one part of the light that is transmission through the canopy and another part that is reflected. from the canopy. The SPAD reads adsorbed red light (at 650 nm) through the leaf blade and compares it with transmittance of near infrared light (at 940 nm) which is not absorbed by leaves. These two radiation balance components are used by the SPAD to obtain relative value which is linked to the chlorophyll content [29]. Notice that all alternative methods result in relative values that should be converted to total N content or N uptake through several correlation equations for fertilization management. Each one of the selected equations brings in some errors such that the output cannot be used to determine N deficiency and other management components above.

### 5. Conclusions

The uniqueness of this application that combines under one roof algorithm to calculate N% on the basis of accumulated knowledge [10,12], a simulation model [19] and sophisticated computing capabilities [4,5] that is using smartphones to produce supportive recommendation for agronomists and growers. The output is the amount of nitrogen fertilizer dose to apply as topdressing to wheat fields. This application is the only proximal sensor that offers a deterministic solution of nitrogen fertilization policy for spring wheat.

Summing up the many publications that are offered by literature, the major difference between this proximal sensing and the others is that while all others are based on combination of fertilization indices and agronomic intuitions the output of this study is a real time quantitative support system based on most modern computing software.

It can also be mentioned that the best "sensor" for nitrogen management is the plant itself while the cellular phone is a "data logger" for the plant signals. Thus, the study offers inexpensive and readily available smartphone to collect the data while making real time N management decisions. Other different sensors may improve nitrogen fertilization assessment but they are less available than smartphones.

RGB wide bands can be used to estimate N% status of wheat with $r^2 = 0.98$ and compared to laboratory measurements accuracy of more than 80% (Figures 3 and 4A,B). Farms that could benefit from this nitrogen study are farms where little or no nitrogen is applied to wheat or other crops. Farms with production of 2.75 Mg·ha.$^{-1}$, which is the official lowest recorded wheat yield or farms with production less than 4.4 Mg·ha.$^{-1}$, the world average production, can most benefit from this study.

Difficulties involved in the implementation of proximal sensing include (a) calibration requirements. For the wheat calibration we used about one hundred samples for laboratory comparison. Currently there is no universal sampling solution for all crops. Each crop would need its specific calibration procedure. (b) Nitrogen uptake was associated with collecting canopy dry matter which is associated with time and labor requirements. Alternatively crop growth simulation model for wheat (DSSAT) was used. Disturbances that were treated through the calibration process with machine learning and ANN were solved by noon time sampling to avoid morning limited sun light or cloudiness and varied smartphones were tested (5–6 different cameras) and found with negligible effects.

**Author Contributions:** Conceptualization J.B.-A., A.Z., and I.R., Formal analysis E.S., S.S., Investigation I.R., data curation S.N., M.D., Software Writing A.Z., —original draft J.B.-A. All authors have read and agreed to the published version of the manuscript.

**Funding:** This study was financially supported by the Ministry of Science and technology Israel AgrIOT.group LTD and Katif R&D center for coastal deserts development. The Field Crops Organization. Ministry of Agriculture Israel.

**Conflicts of Interest:** The authors declare no conflict of interest.

## Abbreviations

| | |
|---|---|
| ANN | Artificial Neural Network (dimless) |
| DAE | Day After Emergence (days) |
| DM | Dry Matter ($Kg\ ha^{-1}$) |
| DSSAT | Decision Support System for Agricultural Technology |
| DSS | Decision Support System |
| HI | Harvest Index (dimless) |
| Nc | Critical Nitrogen ($g\ kg^{-1}$) |
| NNI | Nitrogen Nutritional Index (dimless) |
| RE | Relative Error (dimless) |
| RGB | Red Green Blue (wave bands) |
| RMSE | Root Mean Square Error |
| STDEV | Standard deviation |

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
