# Peer review of "Proximal Sensing of Nitrogen Needs by Spring Wheat"

_agronomy, doi:10.3390/agronomy11030437_

Round 1

Reviewer 1 Report

The aim of the study in “Proximal sensing of nitrogen needs by spring wheat” was To establish algorithm for using RGB photographs taken by a smart phone in order to determine %N and two major manage ment indices NNI and Nc of wheat and to combine them for fertilization management. The article requires revision before accepted for publication. The specific comments are given below.

  1. Remove abbreviations from the summary. Explain in the text.
  2. Correct the punctuation, e.g. lines 45, 48, 49, 96, 100, 102, 103…
  3. Ln 21-22 “Maximum N uptake calculated from photo and laboratory was 324 kgha-1 and 318 kg ha.-1 respectively” - Pay attention to the units.
  4. Correct the numbering, e.g. 1.1. Proximal sensing of Nitrogen and 1.1. Main goal.
  5. What methods were used to determine the statistical significance of differences between the analyzed variables? What level of significance was adopted?
  6. Standardize the units throughout the manuscript e.g. you use kgha-1 and kg/ha.
  7. Include standard deviations in the tables and figures.
  8. Correct the scale in the figures.
  9. Expand the discussion significantly. Compare and describe the accurate results other researchers have obtained.
  10. Update the references. Cite more recent research.
  11. Provide the most important numerical values in your conclusions.
  12. Remove citation from conclusions. This should be in the discussion.
  13. Add „Abbreviations” before „References”.

Author Response

  1. V Remove abbreviations from the summary. Explain in the text.
  2. V Correct the punctuation, e.g. lines 45, 48, 49, 96, 100, 102, 103…
  3. V Ln 21-22 “Maximum N uptake calculated from photo and laboratory was 324 kgha-1 and 318 kg ha.-1 respectively” - Pay attention to the units.
  4. V Correct the numbering, e.g. 1.1. Proximal sensing of Nitrogen and 1.1. Main goal.
  5. V line 294 What methods were used to determine the statistical significance of differences between the analyzed variables? What level of significance was adopted?
  6. V Standardize the units throughout the manuscript e.g. you use kgha-1 and kgha-1.
  7.  
  8. V Include standard deviations in the tables and figures.

STDev was included to tables 5.and 6. In fig 1 we included statistical confidence limit. 

  1. V Correct the scale in the figures.
  2. V Expand the discussion significantly. Compare and describe the accurate results other researchers have obtained.
  3. V Update the references. Cite more recent research.

V Provide the most important numerical values in your conclusions.

  1. V Remove citation from conclusions. This should be in the discussion.

V Add „Abbreviations” before „References”.

Reviewer 2 Report

The authors presented a method based on using RGB images to determine the nitrogen (N) content for spring wheat and they further combine it with a modified N dilution curve to estimate the total N requirements. The results are interesting, however, a detailed description of the methodology used is needed.

As was stated in the paper, deep learning was used to translate RGB images of the canopy into N concentration. So, the reader would expect to find a clear description of the methodology used which is missing. It is not clear which feature(s) were extracted from the RGB images and used during the training process. It is also not clear whether the authors considered a validation step. But it seems from the manuscript that the same data was used all along without considering a separate data set for the validation step. The main drawback is therefore the lack of a validation step which needs to be addressed.

The same applies to the modified N dilution curve. The argument to substitute DM by DAE was the fact that they are linearly correlated as stated in line 303. However, this linear relationship is not clear and it seems to be cultivar dependent (Fig. 5). It would be helpful to add a figure showing the DM vs DAE and whether the same relationship holds for both the greenhouse and field growing conditions. Moreover, with the modified N dilution curve, three parameters need to be fitted (i.e. Ncmax, DAE50 and b). So, how Ncmax is determined? It seems from Fig. 3 that Ncmax is measured. Maybe the authors can describe how %N was determined at DAE=0 which is missing in the methodology section. How DAE50 can be determined in practice especially that it would highly depend on the growing conditions (warm season).

The discussion needs to be improved and as said more details are needed for the methodology used to develop the algorithm to translate the RGB images into %N.

Author Response

The authors presented a method based on using RGB images to determine the nitrogen (N) content for spring wheat and they further combine it with a modified N dilution curve to estimate the total N requirements. The results are interesting, however, a detailed description of the methodology used is needed.

Our  methodology was described in details in proceeding papers No [4 & 5] This point is emphasized strongly in the text

As was stated in the paper, deep learning was used to translate RGB images of the canopy into N concentration. So, the reader would expect to find a clear description of the methodology used which is missing. It is not clear which feature(s) were extracted from the RGB images and used during the training process. It is also not clear whether the authors considered a validation step. But it seems from the manuscript that the same data was used all along without considering a separate data set for the validation step. The main drawback is therefore the lack of a validation step which needs to be addressed.

The adaptation of the method include the following stage 1. Calibration of about 70% of the 100 samples . 2. Validation on the field 20-30% of the samples and 3. Independent field validation.

The data presented here include calibration and validation. The point was addressed in  lines 175-179

Photographs and explanations of the process were added in line 77-88

The same applies to the modified N dilution curve. The argument to substitute DM by DAE was the fact that they are linearly correlated as stated in line 303. However, this linear relationship is not clear and it seems to be cultivar dependent (Fig. 5). It would be helpful to add a figure showing the DM vs DAE and whether the same relationship holds for both the greenhouse and field growing conditions. Moreover, with the modified N dilution curve, three parameters need to be fitted (i.e. Ncmax, DAE50 and b). So, how Ncmax is determined? It seems from Fig. 3 that Ncmax is measured. Maybe the authors can describe how %N was determined at DAE=0 which is missing in the methodology section. How DAE50 can be determined in practice especially that it would highly depend on the growing conditions (warm season).

The four points in each treatment at fig 5 are given as a polynomial graphs with r2≈1 but testing   them on a linear curve fit showed that linearity can be approximated with r2 varying from 0.86 to 0.98. It is different for each variety , treatments and growth conditions. The point was addressed in the discussion lines 615-622 with additional citation.

). So, how Ncmax is determined?  See lines 327-330

The discussion needs to be improved and as said more details are needed for the methodology used to develop the algorithm to translate the RGB images into %N.

The discussion has been improved and extended accordingly. It  highlighted with a yellow mark on the revised  paper

Reviewer 3 Report

Please re-write the abstract to be more informative. Now is just the result report.

Line 48: remove,

Line 142: the first time “Deep Leaning”!

Please add a flowchart to show the steps clearly.

Please explain the calibration, training, and validation steps in machine learning.

Please compare the results with the other works.

Please explain the deep learning model in detail. Hyperparameters?

Did you use the images as inputs? Or signals? Please show some of them.

Author Response

Yes

Can be improved

Must be improved

Not applicable

Does the introduction provide sufficient background and include all relevant references?

( )

(x)Improved

( )

( )

Is the research design appropriate?

( )

(x)

( )

( )

Are the methods adequately described?

( )

(x)

( )

( )

Are the results clearly presented?

( )

(x) Improved

( )

( )

Are the conclusions supported by the results?

( )

(x) Improved

( )

( )

Comments and Suggestions for Authors

Please re-write the abstract to be more informative. Now is just the result report. Abstract was rewritten

Line 48: remove,

Line 142: the first time “Deep Leaning”! Changed to Machine learning

Please add a flowchart to show the steps clearly Photographs and explanations of the process were added in line 77-88

Please explain the calibration, training, and validation steps in machine learning. The point was addressed in  lines 175-179 and in citation papers  [4&5]

Please compare the results with the other works.

Please explain the deep learning model in detail. Hyper parameters? in citation papers  [4&5]Ziberman and Ben-Asher ,  two of the leading authors addressed this point including a flowchart

Did you use the images as inputs? Or signals? Please show some of them.

We are using images and schematically the are displayed in Photo 1 line 77-88

Round 2

Reviewer 1 Report

Thank you for considering my comments.

Author Response

1

Reviewer 2 Report

I appreciate the great efforts that the authors have made in response to my questions and concerns. However, the answers they provided and the changes they have made to the text just made things even more confusing. 

My main concern is the use of DAE as a proxy for biomass and the explanation was that the relationship between DAE and biomass was linear. In their response, they already acknowledge that this relationship differs for different growing conditions which is expected. Therefore, it is not appropriate to use DAE, using GDD would have been more suitable.

Besides, it is not clear how DSSAT was used. Line 693-695: "Currently, the model is an integrated algorithm that include %N from RGB photograph , Nc and NNI and ΔNNI from Eqs 5, 6 and 7 and DSSAT output of daily top weight for wheat." This means that DSSAT is part of the method they developed. However, there is no indication whether DSSAT was calibrated or not and how growers are going to use it. Moreover, if DSSAT is used then what is the point of substituting the biomass by DAE! Why not just use DSSAT to get the amount of N needed?

The introduction does not provide enough background and therefore needs to be revised along with the objectives. Even though part of the developed method was described in some of the previous work, I think it would be helpful to include a short description in the manuscript to help readers understand the current manuscript. The discussion was not improved. So, the authors still need to discuss their results and link their discussions to findings from the literature. Fig 1 is different from the first version and so is the linear equation relating the N uptake determined by the camera and in the laboratory. 

I highly encourage the authors to take enough time to revise the paper as it requires significant editing.

Author Response

1

Reviewer 3 Report

The authors answered all my questions. The only remaining issue is:

please add a flowchart to the paper. 

Author Response

1